# The Use of Lupin as a Source of Protein in Animal Feeding: Genomic Tools and Breeding Approaches

**DOI:** 10.3390/ijms20040851

**Published:** 2019-02-15

**Authors:** Eleni M. Abraham, Ioannis Ganopoulos, Panagiotis Madesis, Athanasios Mavromatis, Photini Mylona, Irini Nianiou-Obeidat, Zoi Parissi, Alexios Polidoros, Eleni Tani, Dimitrios Vlachostergios

**Affiliations:** 1Laboratory of Range Science, School of Agriculture, Forestry and Natural Environment, Aristotle University of Thessaloniki, 54124 Thessaloniki, Greece; pz@for.auth.gr; 2Institute of Plant Breeding and Genetic Resources, HAO-DEMETER, Thermi, 57001 Thessaloniki, Greece; giannis.ganopoulos@gmail.com (I.G.); phmylona@nagref.gr (P.M.); 3Institute of Applied Biosciences, CERTH, 57001 Thessaloniki, Greece; pmadesis@certh.gr; 4Laboratory of Genetics and Plant Breeding, School of Agriculture, Aristotle University of Thessaloniki, 54124 Thessaloniki, Greece; amavromat@agro.auth.gr (A.M.); nianiou@agro.auth.gr (I.N.-O.); palexios@agro.auth.gr (A.P.); 5Department of Crop Science, Laboratory of Plant Breeding and Biometry, Agricultural University of Athens, Iera Odos 75, 11855 Athens, Greece; 6Institute of Industrial and Forage Crops, HAO-DEMETER, 41335 Larissa, Greece; vlachostergios@gmail.com

**Keywords:** lupin, animal nutrition, genomics era, breeding, molecular markers, variety development

## Abstract

Livestock production in the European Union EU is highly dependent on imported soybean, exposing the livestock farming system to risks related to the global trade of soybean. Lupin species could be a realistic sustainable alternative source of protein for animal feeding. *Lupinus* is a very diverse genus with many species. However, only four of them—namely, *L. albus*, *L. angustifolius*, *L. luteus* and *L. mutabilis*—are cultivated. Their use in livestock farming systems has many advantages in relation to economic and environmental impact. Generally, lupin grains are characterized by high protein content, while their oil content is relatively low but of high quality. On the other hand, the presence of quinolizidine alkaloids and their specific carbohydrate composition are the main antinutritional factors that prevent their use in animal feeding. This research is mainly related to *L. albus* and to *L. angustifolius*, and to a lesser extent, to *L. lauteus* and *L. mutabilis.* The breeding efforts are mostly focused on yield stabilization, resistance to biotic and abiotic stresses, biochemical structure associated with seed quality and late maturing. Progress is made in improving lupin with respect to the seed quality, as well as the tolerance to biotic and abiotic stress. It has to be noted that modern cultivars, mostly of *L. albus* and *L. angustifolius*, contain low levels of alkaloids. However, for future breeding efforts, the implementation of marker-assisted selection and the available genomic tools is of great importance.

## 1. Introduction

Animal products are the main source of protein for European citizens, with 59% of the consumed protein in the European Union (EU) being derived from them [1]. Animal feed is mainly based on plant proteins, animal by-products (residue materials, e.g., blood, bones, feathers, hides and skins) and fishmeals. The use of animal by-products in livestock feeding is very limited in the EU due to EU legislation for safety [2]. Thus, the livestock sector is by far the largest market for plant proteins in the EU [3]. Among the plant proteins, soybean is the most commonly used in animal nutrition. This is because of its high protein and amino acid contents, as well as its availability throughout the year.

Generally, the EU is self-sufficient with respect to meat production, with 96% of the available meat being produced in EU countries [1]. Conversely, EU has a deficit in plant protein production, especially in soybean. Only 5% of soybean demand was produced in EU countries in 2017 [4], making the EU the largest soybean importer [5]. Consequently, livestock production in the EU is highly dependent on imported soybean. This dependency is higher for monogastric livestock (pigs and poultry) compared to ruminants [6]. This dependency exposes the livestock farming system of the EU to risks that are related to the global trade of soybean. The expected increase in meat and milk consumption internationally by 68% and 57%, respectively, by 2030 [7] and, as a consequence, the requirements for animal feeds, is likely to contribute to the volatility of soybean availability and prices.

The EU intends to address the imbalance between supply and demand for plant proteins by promoting European protein crops [4]. Measures of the recent Common Agricultural Policy (CAP), which support protein crops, have already contributed to the expansion of plant protein sector in EU. The above political decision was dictated by financial, environmental, social and product quality issues. In recent years, consumers have been becoming more and more aware of issues concerning animal products, such as animal welfare, animal nutrition (organic or non-genetically modified feed non-GM) and environmental impact. This trend is more obvious in non-GM-fed animal products. For example, only 9% of cattle feed was non-GM in Germany in 2012, while in 2017 it was more than 40% [4]. Furthermore, organic animal production in the EU is constantly growing. The use of genetically modified feeds is not allowed in organic farming. However, the availability of non-GM soybean is very low, and the price is very high [8]. In this regard, finding alternative sources of protein for animal feed is imperative. Additionally, the dependence of livestock production on the international soybean market has led to an increase in the use of locally produced feed such as grain legumes.

Grain legumes or pulses are an important element of sustainable agricultural production, human nutrition and livestock feeding. Symbiotically with *Rhizobium* bacteria, they fix atmospheric nitrogen, and therefore, their cultivation does not require the use of inorganic nitrogen fertilizers. Their use in intensive rotation system with cereals improves the yield production of cereals and mitigates environmental impacts such as demand for non-renewable energy resources, eutrophication, acidification, terrestrial and aquatic ecotoxicity [9]. Moreover, they can be used in the diet of ruminants either as concentrate or as forage. Replacement of soybean with grain legumes in ruminant diets such as *Cicer arietinum*, *Pisum sativum* and *Vicia faba* without negative impact on the quantity and quality of animal products has been reported in many research studies [10]. On the other hand, their relatively high content of secondary metabolites (anti-nutritional factors) may prevent their use in animal feed, especially in that of monogastric livestock, which are generally more susceptible to most anti-nutritional factors compared to ruminants [11].

Among the grain legumes, lupin species seem to be a realistic sustainable alternative source of protein in both monogastric and ruminant feed that can replace soybean without loss in the quantity and quality of the livestock products [12,13]. Nevertheless, nutritional value of lupin varies greatly among varieties, and breeding efforts have so far been limited, indicating great potential for further improvement. It should be noted that efforts towards lupin breeding have been launched recently compared to all the other food crops. The aim of the present study is to review the use of lupin species in animal nutrition. The existing information on their nutritional value and anti-nutritional factors is presented. Emphasis is placed on the available genomic resources, conventional breeding for the release of new cultivars, and also on the use of genomic tools in breeding programs.

## 2. Economic Importance, Growing Region

### 2.1. The Genus Lupinus

*Lupinus* is a very diverse, widespread genus of the Fabaceae family with numerous species. It is distributed in a wide range of climatic conditions, from the subarctic region to semi-desert and subtropical climates, as well as from sea level to alpine ecosystems (4000 m altitude). The species of the genus can be categorized into two groups: (a) “Old World” species, Mediterranean, North and Eastern African; and (b) “New World” species, North and South America [14]. The “Old World” group consists of only 12 annual species and is divided into: (a) Malacospermae, smooth-seeded species of *L. angustifolius*, *L. albus*, *L. luteus*, *L. hispanicus* and *L. micranthus* with chromosome number from 2*n* = 40 to 52; and (b) Scabrispermae, rough-seeded species of *L. pilosus*, *L.cosentinii*, *L. digitalis*, *L. prinei*, *L. palaestinus*, *L. atlanticus* and *L. somaliensis* with chromosome number from 2*n* = 32 to 42 [15]. On the other hand, the “New World” group consists of many more species that are annual and herbaceous perennial, as well as a few of them that are shrubs [16]. The estimated number of “New World” species is approximately 280 but they are not well taxonomically defined [17]. The phylogenetic relationship among the *Lupinus* species is reviewed in detail by Wolko et al. [18]. However, among the numerous species of the genus, only *L. angustifolius* (blue lupin or narrow-leaved lupin), *L. albus* (white lupin), *L. luteus* (yellow lupin) from the “Old World” group and *L. mutabilis* (Andean lupin) from “New World” group are cultivated.

### 2.2. The Uses and Production of Lupins

The cultivation of lupins has a long history extendin back to ancient times. *L. albus* was cultivated in ancient Greece, Egypt, Rome and other Mediterranean countries before 2000 BC for green manure, animal and human consumption, as well as for cosmetic and medicine use. The domestication of *L. mutabilis* took place in the Andean states in about 700–600 BC, while that of *L. luteus* and *L. angustifolius* in Baltic countries in the 1860s [19,20].

Lupins are mostly used as fodder and food crops, and some species are also used ornamentally. They are traditionally part of the human diet mainly in the Mediterranean region and in the Andean highlands of South America [21], although this use is limited (only 4% of global production) [22]. However, recent reports about the nutritional properties of lupins have enhanced their use in a variety of functional foods [23]. Additionally, lupins as legumes fix atmospheric nitrogen, contributing to the enhancement of soil fertility, and also the yields of the subsequent crops in rotation systems [24]. The wheat:lupin rotation system has been used for more than 40 years in Western Australia for sustainable wheat production, while the lupin stubble residues are grazed by livestock [25]. Nevertheless, the commercial value of lupins is primarily based on lupin seed production as stock feed for ruminant and monogastric livestock, as well as for aquaculture [26].

Lupin production and cultivated area worldwide for 2017 is estimated at about 1,610,969 tonnes and 930,717 ha respectively [27]. Oceania is the largest producer, with 64% and 55% of the global production and cultivated area, respectively (Figure 1), while Europe is second. *L. angustifolius* is the main cultivated species in Australia [26] and northern Europe, while *L. albus* is the dominant one in southern Europe [24]. The percentage of global production attributed to Europe increased remarkably from 17.6% in 2013 [28] to 29% in 2017 (Figure 1). However, the cultivation of lupins in Europe is still limited because of the low number of target breeding programs, which is a crucial factor for low production and low expansion of lupin cultivation in Europe, as it has exploited little of the gene pool [29]. Identification of germplasm with tolerance to a range of abiotic stresses (calcareous soils, drought, occasional frost, etc.) is expected to expand lupin cultivation into a wider range of agro-climatic conditions across Europe, and it may increase grain or biomass production. Another key issue is the development of adaptive *Bradyrhizobium* strains that promote nodulation under various stresses and result in better plant growth and yield performance [30]. Furthermore, low productivity due to seasonal variability [31], low price of lupin grain [1] and EU policies that favored the importation of soya bean contributed to the decline of lupin production in Europe, especially during the second half of the 20th century [28,32]. Pulse cultivation occupied only 2.1% of the arable land in Europe in 2016, and lupins were grown on 11.9% of this area [33]. According to Lucas et al. [28], lupins could be a primal cultivation in different agro-climatic zones and marginal lands in Europe.

## 3. The Use of Lupins in Livestock Farming Systems

### 3.1. Seed Yield Production

Lupins have been suggested as alternative protein crops in Europe, as they offer the possibility of reducing the quantities of imported soya bean meal, replacing it with a high-quality traceable source of animal feed protein [1]. However, the varieties that are currently cultivated are characterized by relatively low and unstable seed yield. One of the best cases is winter hardy autumn sown varieties, which are capable of producing from 2.5 to 4.0 t/ha, a quantity near to the level of soybean production (3 to 4.5 t/ha) [34].

The factors that influence the total yield are related: (1) To the area of cultivation, because it is necessary to avoid soils with high pH (above 7.0) and soils vulnerable to water logging. (2) The sowing period (from the end of September until end of January) and seed density (from 30 to 45 seeds/m or 85–120) regulated for achieving maximum yield. Improved varieties which mature earlier are less dependent on early sowing and are more tolerant to high soil pH values [35].

The comparison with soybean is not an objective that could be fairly judged, as lupins are crops adapted to low input farming, and soybean is a spring crop with high water demands. Thus, the critical point is the price of seeds per tonne or the price of mill per tonne. Lupins give an income of around £100/tonne (120 euros/tonne), which is roughly the same as the income derived from soybean [35], if we assume that soybean is sold as treated mill [34]. In terms of research on yield-related traits, it must be underlined that the main target of lupin breeding for many decades has been the reduction of the alkaloid levels in the seed, and secondly the yield or yield components.

### 3.2. Nutritive Value and Antinutritional Factors

The nutritive value of lupin grain is highly depended on species, genotypes and location. There are reports [36,37,38,39] on the effect of environmental conditions such as temperature and soil properties on seed quality traits. Generally, all the cultivated species have relatively high protein content (Table 1) comparable to soybean. Higher protein content was recorded for *L. mutabilis* and *L. luteus* compared to the others. However, research related to the use of both of them in animal feeding is very limited. Additionally, lupin grain contains essential amino acids in higher levels than those of soybean [40]. However, lupins’ grain has lower content of the nutritionally essential, sulfur-containing amino acid than soybean, and this is a disadvantage in their use as feed of monogastrics [41].

The oil content of lupins is generally quite low compared to oilseeds, apart from *L. mutabilis* (Table 1), but with a high concentration of polyunsaturated fatty acid [42]. However, the quality of oils in terms of unsaturated fatty acids (UFAs)/Saturated fatty acid (SFAs) and n-3/n-6 polyunsaturated fatty acids (PUFA) ratio is more important than the quantity from the nutritional point of view [43]. In this regard, Chiofalo et al. [44] described a high-quality oil profile for the grains of both *L. albus* and *L. luteus* in comparison to other vegetable oils, and they suggested their use as a nutraceutical feed in order to enhance the nutritional value of animal products.

The presence of quinolizidine alkaloids provides the bitter taste of lupin grain. The alkaloid profile is species specific, and its level is generally higher in grains compared to the vegetative parts of the plants [18]. However, the alkaloid level of current cultivars is very low [51] and does not affect feed intake. On the other hand, anti-nutritional factors such as trypsin inhibitors and saponins are very limited in lupin grain [42,52].

The main anti-nutritional factor of lupin grain is related to their specific carbohydrate composition [13], which is characterized by low levels of starch, high levels of Non-Starch Polysccharides (NSP) [53,54], and high levels of raffinose oligosaccharides [18]. These properties affect the utilization of energy and contribute to the reduction of feed intake and digestibility, mainly in monogastrics. In addition to breeding efforts, mechanical and biological processing methods such as grinding, soaking, heating, etc., are used in order to eliminate the anti-nutritional factors of lupin grain [55].

### 3.3. The Lupins in Livestock Farming Systems

The use of lupins in livestock farming systems for both ruminants and monogastrics has many advantages. They can be used in animal diets either as concentrate (whole seeds, ground seeds or other processed seeds) [12] or as forage (whole-crop, silage or hay) [56]. Beyond the nutritional value, lupins are characterized by high grain productivity [44], they are adapted to poor and barren soils [42], they have fewer requirements than other crops [57], and finally, they are an excellent rotation crop [58]. Furthermore, White [13] suggested that dairy farmers in Australia prefer to use lupins as a supplementary feed source because they are generally cheaper than oilseed proteins, and they are easy to store and handle. According to Chiofalo et al. [44], *L. albus* is suitable for the Mediterranean crop-livestock food chain, while *L. luteus* is a promising lupin species for the same use.

Lupins are an excellent source of energy and protein, and can effectively replace soybean in ruminant feed. White [13] extensively reviewed the use of lupins as a feed for dairy cows in Australia. According to this review, the production of milk, fat and protein increased and was not affected by the use of lupins instead of cereal grains and soybean, respectively. However, the use of lupins improved the fatty acid profile of milk in relation to human health. Similarly, soybean meal was efficiently replaced by lupin seeds in diet of high-producing dairy cows [59], of Podolian young bulls [60], of lactation ewes [61] and of Washera Sheep [62,63].

Unlike ruminants, the use of lupins in the diet of monogastrics could be detrimental for their performance and production. The lack of endogenous enzymes for degradation of lupins’ carbohydrates in their digestive system can reduce feed intake, digestible energy and digestibility of nutrients. However, the use of *L. angustifolius* as a protein source in the diets of laying hens with inclusion of NSP degrading enzymes did not negatively affect the production and health of hens [64]. Similarly, the dehulled-micronized seeds of *Lupinus albus* cv. Multitalia could replace soybean in the diet of early-phase laying hens [65]. On the other hand, the seeds of *L. angustifolius* were quite inferior to those of *Pisum sativum* and of *Vicia faba* as a source of energy in the diet of broilers [66]. Finally, Pierre et al. [55] indicated that high grinding intensity and hydrothermal processing increased the nutritional value of *L. angustifolius* and concluded that it can replace soybean in the diet of growing pigs.

## 4. Available Genomic Resources and Breeding Trends

Genetic resources are the cornerstone of breeding, particularly when addressing current challenges that threaten global food security, including population growth, climate change, reduced water resources for irrigation, soil erosion, and the increasing costs of fertilizers [67,68]. Thus, diversification of modern cropping systems to adapt to these challenges is a major concern. Today, the collection of lupin germplasm encompasses 38,000 accessions, reviewed by Smykal and co-workers [69]. The collection of *ex situ* lupin germplasm is highlighted by the portion of wild species (18%) compared to the cultivated. Due to the species’ late domestication and cultivation, the collection of wild diversity represents a great scientific and breeding focus, which is addressed herein.

The Mediterranean region is considered the origin of lupin species of the “Old World” [70]. The smooth-seeded indigenous lupin species of the Mediterranean diversity include the commonly known white lupin (*L. albus*), the yellow lupin (*L. luteus*) and the blue lupin (*L. angustifolius*), based on the dominant seed color for *L. albus* and the dominant flower color for *L. angustifolius* and *L. luteus*, respectively. Several subspecies of white lupin expand the Mediterranean lupin genetic resources further, including *L. albus*, subsp. graecus, subsp. thermis and subsp. albus, setting the Balkans as a center of lupin diversity. In Mediterranean agriculture, lupin used to be a crop of marginal lands providing protein for both human and animal diet.

Cultivation of white lupin dates back to ancient times in Greece, according to the descriptions of Theofrastos, Ippocrates and Dioskouridis. Due to this early lupin cultivation, recent studies have demonstrated that phenology in all three lupin species has been under strong selection along the arid gradient zone [71]. While narrow-leaved lupin (NLL) domestication was only completed in the twentieth century, there is ample evidence of its use by Mediterranean cultures for centuries [72].

Blue lupin (*L. angustifolius*), the wild-grown narrow-leaved lupin of the Mediterranean, has a shorter and fragmented domestication history in comparison to its white and yellow counterparts, as its use was restricted to forage crops and green manure, owing to its high content of bitter-tasting alkaloids. Ιt should be noted that aside from the three aforementioned lupin species, several other “Old World” lupin species have been used sporadically in agriculture, including the rough-seeded *L. pilosus*, *L. atlanticus* and *L. cosentinii* [18,72]. Intercrossing among lupin species, including the main cultivated species, is rarely successful [18]. Thus, domestication and cultivation of lupin species has largely been hindered by their alkaloid content.

Consequently, the major breeding thrust in the mid-1920s and 1930s centered on alkaloid-free lupin, promoting lupin cultivation in Europe and around the globe. Alkaloid-free lupin was achieved by a set of simple inherited natural and spontaneous mutant genes that reduce the alkaloid content in the three cultivated lupin species of Europe [72]. However, cultivation of lupin was still delayed in Northern climates by the inability of NLL to ripen in a timely manner, until the removal of vernalization response. The latter is controlled by a mutation in a single gene based on the recent insight gained by Nelson and co-workers [73]. Specifically, it was shown that the time of flowering of NLL was mediated by its responsiveness to vernalization (cold conditions), day length and ambient temperature via the *Flowering locus T* (FT) gene. Thus, a mutation in this locus obtained during breeding efforts led to cultivation attainment beyond the temperate Mediterranean region.

Given the difficulty of interspecific crossing between the cultivated species of lupin, no example of successful gene transfer of quantifiable trait among lupin crop species has been documented, thus far. Therefore, for practical breeding purposes, the gene pool of the cultivated “Old World” lupin species is rather restricted to the species themselves. Yet, the Mediterranean lupin diversity remains untapped, with valuable genetic traits that could be harnessed to address the major constrains of agriculture, including drought and climate change. In this respect, the latest study of the migration history of NLL, coupled to identification of genome regions associated with adaptation, highlighted the importance of resourceful adaptation traits to the warmer and drier climate of the eastern Mediterranean. Hence the discovery of earlier flowering, as natural selection, was inevitably used as an escape strategy in eastern Mediterranean populations for adaptation to the drought and heat stress. In this regard, evolutionary studies may pinpoint genes that natural selection may now be sorting out of the gene pool.

Thus, the genetic resources of lupin populations of the “Old World” include a diversity of eco-geographically adapted populations; outcomes of natural selection that remain unexplored. With the advancement of high-throughput sequencing technologies and the availability of computational platforms and databases, a detailed assessment of interesting traits and genetic regions across lupin genomes, along with geo-climatic data, could provide valuable insights into variants and genetic loci. This knowledge will enable specific targeting of germplasm for screening of traits including tolerance to abiotic and biotic stresses for the development of lupin cultivars with greater resilience to the environment under climate change.

Apart from the three “Old World” cultivated lupin species, the “New World” species *L. mutabilis* encompasses a diversity of agronomically important traits. The *L. mutabilis* germplasm is apparently under robust exploration, as the species possesses the highest grain quality of all the cultivated lupins, matching soybean, with an average of 42% protein, 18% oil content and a thin seed coat [18]. Along with its tolerance to water logging and its high P-use efficiency [18], the species breeding in the 1940s was centered on low alkaloid content. This target was achieved by new cultivars with <0.05% alkaloids [74]. Yet, *L. mutabilis* implementation in cropping systems is limited due to the late maturity, low and unstable yields, and frost susceptibility [75].

Conclusively, the breeding aim is centered on broadening the genetic base by combining distantly related wild and domestic material with elite cultivars. Nonetheless, this process could be accelerated today by marker-based analyses of genetic diversity in geographically defined germplasm sets of cultivated lupine species. Consequently, systematic investigations that integrate both molecular and ecophysiological information are required for both the “Old World” and the “New World” cultivated species to targeted trait introgression to improve lupin adaptation, quality/value, and cropping fitness in a climate changing environment.

## 5. Breeding of Lupins

### 5.1. Target Traits for Breeding and Cultivar Release

White, yellow, and narrow-leaved lupins are native European legumes that can become alternatives to soya bean, given their elevated and high-quality protein content, potential health benefits, suitability for sustainable production, and acceptability to European consumers [28]. During the last century, breeding programs in lupins achieved to increase seed yield along with the improvement of significant agronomical and seed quality traits [76]. A basic group of traits that were bred and which contributed to the domestication and improvement of the lupine species through the years consisted of high water permeability of seeds, non-dehiscing pods, flower, cotyledon and seed pigmentation control, improvement of harvest index, restricted branching, and duration of vegetation (see [76] and references therein).

Modern breeding programs mainly focus on yield stabilization, resistance to abiotic stresses, biochemical structure associated with seed quality, resistance to diseases (mainly to anthracnose) and late maturing [30,77,78].

Another interesting trait for breeders is seed oil content (8–14% in white lupin of excellent nutritional quality) [79]. Given the high protein of lupin seeds, further genetic improvement of oil content would increase the economic sustainability of the crop by making it dual-purpose (protein and oil), like soybean [80].

The sources of genetic variation used to achieve the breeding targets were primarily landraces and new variants derived after mutagenesis [14,81]. A typical successful example was the 10-year breeding program in Australia for the development of *L. albus* varieties with resistance to anthracnose that used landraces from Ethiopia as a source of genetic variability [82]. In this program, landraces were screened for valuable resistance *to Colletotrichum lupini*, and subsequently, crosses between the resistant landraces and agronomically advanced lines were implemented. Finally, the anthracnose-resistant variety “Andromeda” was released for commercial production [82,83]. Similarly, Jacob et al. [84] used gene bank accessions to apply numerous crosses with a commercial variety and after multi-environmental trials they produced lines that combined superior yield, low alkaloids and resistance to anthracnose.

On the other hand, mutagenesis proved to be a useful tool to generate novel genetic variation and the majority of lupin cultivars were produced by the use of spontaneous or induced mutants [81]. The first sweet fodder cultivars of yellow, narrow-leaved and white lupin were bred in Germany in 1927–1928 as a result of reproduction from an identified mutant plant with low alkaloid content [85]. In addition, several mutants were produced through irradiation and were used for the development of low-alkaloid varieties, such as Kiev Mutant [86]. Furthermore, other mutagens were used to exploit genes for early flowering that were finally implemented in *L. angustifolius* breeding [81], as well as mutants giving resistance to the herbicide metribuzin [87]. Natural mutants were used in hybridization with commercial cultivars for improvement of many parameters (white coloring and fast swelling of seed, non-dehiscence and non-pubescence of pods, fast rates of initial growth using pedigree (PD.M) and single seed descent (SSD) methods [81,88].

Gene introduction from wild relatives is also an alternative source of genetic variability, although produced lines are usually of inferior agronomic quality because unwanted traits are tightly linked to the traits of interest [89]. However, wild relative collections are considered to be of significant value, as resistant genes are more frequently found [29].

A challenging issue in lupin breeding programs is the selection for two or more traits from genetic resources and their transfer to cultivars [42,81]. In this case, the correlation pattern between the target traits is the crucial point. For example, a strong negative correlation between anthracnose resistance and late maturing was observed [83]. Genotypes with anthracnose resistance and late flowering cannot produce pods and fill seeds at the end of the season [82]. Moreover, the relationships among lupin traits are significantly influenced by the fluctuating conditions of the growing environments [90,91]. Thus, genotype selection should be based on multiple trait data in variable environments within the target regions [78].

Future progress is expected to be-based in implementing marker-assisted selection or genomic selection for improving complex quantitatively inherited traits using a combination of phenotyping and genotypic data [92,93,94,95].

### 5.2. Progress and Prospects in Lupin Breeding for Biotic Stresses

Biotic and abiotic stresses cause severe losses in plant production, and this is the case for legumes as well. The main biotic stresses with respect to legumes have fungal origins; however, insects, nematodes, viruses, bacteria and parasitic weeds can also infect legumes and reduce their yield [96]. Stability of lupin seed production will be improved by the development of disease-resistant lupin cultivars.

Although lupin does not suffer from many diseases, there are a few that cause high yield losses. The major fungal diseases are Fusarium wilt and anthracnose caused by *Colletotrichum lupinei* [97,98]. Anthracnose became one of the most significant factors limiting lupin development and production, even causing losses of up to 100% [97,99].

The first incidence of anthracnose in lupin was reported in the USA in 1939 on narrow-leaf lupin, and was later assigned to *C. acutatum* [100]. However, Nirenberg et al. [97] reported that lupin anthracnose is caused by a separate species: *C. lupine*. Genetic characterization of *Colletotrichum* isolates related to anthracnose of lupins revealed low interspecificity [101]. The best practice in terms of environmental and economic impact in order to counteract diseases is the development and use of disease-resistant (or tolerant) cultivars [102].

In Australia, a major cultivar used for breeding anthracnose-resistant varieties is ‘Tanji’, whereby its resistance is controlled by a single dominant gene *Lanr1* [103]. Marker-assisted selection using the AnManM1 marker (later improved using Restriction-site Associated DNA (RAD) markers which are tightly linked to the *Larn1 gene* [103]), facilitated the development of resistant cultivars [104]. A step forward has been made in implementation of marker-assisted selection, in terms of anthracnose resistance in lupin breeding programs for various *Lupinus* spp. [92,94]. Recently, other resources of resistance were identified based on European-bred germplasm [92], as well as in white lupin from Ethiopian germplasm [105]. The above research facilitated the development of the first anthracnose-resistant *L. albus* cultivar ‘Andromeda’. This cultivar was the result of an F3 generation-derived single-plant selection of a cross between an Ethiopian anthracnose-resistant landrace P27175 and a well-adapted but highly susceptible breeding line 89B10A-14 from Western Australia [82].

It is expected that next generation sequencing of *C. lupini*, and lupin species will guide the research on anthracnose resistance [100]. In this regard, whole genome sequencing of *Colletotrichum* species has already unlocked a new era for the study of pathogenesis [106]. The first complete genome of *C. acutatum* has become available [107], and also includes a dataset of the *C. lupini* genome. Next-generation sequencing (NGS) has also been implemented to develop the first draft genome of *L. angustifolius* and especially the anthracnose-resistant cultivar ‘Tanjil’ [104,108,109].

Another important fungal pathogen of lupin is lupin fusarium wilt, which is caused by *Fusarium oxysporum* f. sp. *Lupinei*, a soil-borne pathogen affecting roots, hypocotyls or pods. Fusarium is a serious problem, as it causes massive production losses in seed yield and quality [105]. Random Amplified Polymorphic DNA-Polymerase Chain Reaction (RAPD-PCR) analysis was used to identify the ability of resistance to *F. oxysporum* in three lupin cultivars under Egyptian conditions. The effect of infection was determined on weight values of plant, seeds and number of pods. However, the Dijon-2 cultivar showed the highest capacity for resistance to *F. oxysporum* [110].

Protein analysis using Sodium Dodecyl Sulfate Polyacrylamide Gel Electrophoresis (SDS-PAGE) of lupin seedlings showed that application of the biotic and abiotic inducers enhanced rapid induction of different PR-protein in seedlings upon infection with *Fusarium oxysporum* F. sp. *Lupini*, thus providing a useful strategy for the development of induced tolerance of lupin to *F. oxysporum* [111].

Rahman et al. [112] studied the effects of Fusarium wilt disease on the yield, as well as the relationship between plant resistance and biochemical changes in induced plants. A synopsis of the study refers to the induction of resistance by some biotic and abiotic inducers, mainly *P. flurescens* and KCl that might provide a practical supplement to environmentally friendly disease management of soil-borne pathogens, and especially when they are combined with correct integrated agriculture practices. Infantino et al. [113] discussed the recent advances in conventional and innovative screening methods for resistance to several forms of *F. oxysporum*. Breeding programs have been substantially enhanced by the recent identification of molecular markers that are tightly linked to resistance genes. In this case, it has become possible to perform marker-assisted selection (MAS) which in turn could lead to the development of varieties with multiple disease resistance.

Lupins also suffer from *Pleiochaeta setosa*, a fungus causing brown-leaf spot [42]. While most cultivars of lupin species (*L. angustifolius* and *L. albus)* are sensitive to *P. setosa,* other cultivars of *L. luteus* show a high degree of resistance [114]. The knowledge of the mode of resistance inheritance plays a crucial role in successful breeding. However, it is difficult to achieve progress through breeding due to the involvement of a large number of “small-effect genes”. Nevertheless, it is possible. Preventive measures are also important in this respect.

*Phomopsis* is another important disease, mainly because it produces mycotoxins called phomopsins, which result in animal liver disease known as lupineosis [115]. *Phomopsis leptostromiformis* was acknowledged as the causative agent of stem blight in lupins, especially in *L. luteus.* A cultivar of *L albus* “Ultra” is considered to be being resistant, and more importantly does not produce the mycotoxins [116]. “Gungurru” was the first phomopsis-resistant variety released [117]. Shankar and coworkers [118] found that the resistance could be attributed to two genes. Later investigation of wild accessions showed that the resistant trait was able to be inherited [114].

Regarding bacterial infections, lupins are infested by *Pseudomonas* spp. and *Xanthomonas* spp. [119]. *L. albus* suffers drippy pod caused by a pathovar of *Brenneria quercina* bacteria. Due to the high number of genes involved, breeding for resistance is difficult [120].

It is well known that aphids are considered to be lupins’ worst enemies, by infesting the cultivation and transmitting different viruses such as the Cucumber Mosaic Virus (CMV), which can cause significant yield losses. The use of highly specific screening techniques in combination with the correct agronomy practices has led to the identification of lines with low transmission rates caused by CMV [121]. Another important seed-borne virus in *L. albus*, *L. luteus*, *L. pilosus* and *L. atlanticus* is Bean Yellow Mosaic Virus (BYMV), while the infection of *L. angustifolius* is caused by alternative hosts [122]. Aphid species are considered to be CMV and BYMV carriers, transmitting the viruses in a non-persistent manner [123].

Lupin alkaloids are considered to confer insect resistance. The manipulation with the levels of alkaloids and their distribution in the plant is a key element in breeding programs for aphid-resistance cultivars of lupin [124]. An acceptable level of alkaloids in lupins’ seed is 0.05% and 0.02% for animal feed and human nutrition, respectively [39]. However, the low level of alkaloids in seed and the resistance to aphids are two contradictory traits in lupin breeding. Nevertheless, cultivars of *L. angustifolius* with total seed alkaloids of less than 0.02% and with acceptable resistance to aphids were released by breeders in Australia [125]. In these cultivars, the resistance was more related to specific alkaloid profiles (alkaloid gramine or a gramine analogue) than to total alkaloids level [125]. On the other hand, Philippi et al. [126] demonstrated that a challenge for lupin breeders is to release cultivars that contain an acceptable level of alkaloids in the seed for animal feed and a certain level of specific alkaloids in leaves for tolerance to aphids. The alkaloids in lupins are synthesized mostly in the aerial tissues of the plants, and they are distributed by phloem [127]. In this respect, further research is needed regarding alkaloid profiles and the expression of related genes in order to accurately define the sites, levels and distribution of alkaloids in lupins.

### 5.3. Progress and Prospects in Lupine Breeding for Abiotic Stresses

Exploitation of plasticity for adaptation to environmental changes is of great significance for the development of new varieties. Fortunately, lupins are relatively more tolerant to several abiotic stresses than other legumes, and have a proven potential for resilience on poor or contaminated soils [128,129]. For future purposes, identifying germplasm with tolerance to a range of abiotic stresses related to climatic changes may allow lupin cultivation into a wider range of agro-climatic conditions. Although this review focuses on seed quality, it should mention recent advances in breeding for abiotic stress tolerance. As climate change will enforce the frequency as well as the intensity of combined stresses, it is of great importance to address the issues of climate-resilient lupins.

Despite the fact that lupines demonstrate a plethora of desired traits (i.e., high seed protein, excellent performance in crop rotation), they are sensitive to drought and to dry summers typical of Mediterranean, Western Australia and European climates, especially at two phenological phases: (a) seedling stage (lethal); and (b) flowering stage (flower abortion) or “end-of-season drought” [29]. End-of-season drought or terminal drought is responsible for a great reduction in grain yield [24].

As mentioned in a review paper by Palta et al. [130], narrow-leaved lupin appears to have a curious mix of water use strategies, being very wasteful with water when it is freely available and extremely sensitive to drying soils by closing the stomata well before drainage is an issue. One explanation could be that narrow-leaved lupin is well-adapted to sandy soils, thus utilizing a mix of water use strategies. Improvement of narrow-leaved lupin high stomatal sensitivity to water deficit can be accomplished by the following strategies: (a) accessing previously unavailable water through the development of deeper roots; (b) manipulating the leaf area directly; and (c) upgrading osmotic adjustment. Root architecture plays a pivotal role in nutrient use efficiency, abiotic stress tolerance, and carbon sequestration. Genetic variation in desirable root traits has been found in ten wild genotypes of *L. angustifolius* studied in a semi-hydroponic phenotypic system [131].

Lupin breeders in general have exploited few genetic resources due to the necessity of utilizing germplasm with a low alkaloid content [28]. Thus, drought stress tolerance studies in narrow-leaved lupins have been carried out for many years with not great results possibly due to its narrow genetic base [29]. To create genetic variability plant breeders could utilize maternal effects that, i.e., the “contribution of the maternal parent to a phenotype of its offspring beyond the equal chromosomal contribution expected from each parent” as stated by Roach and coworkers [132]. Their results revealed that maternal effects improved drought tolerance by stabilizing seed yield possibly by altering the seed weight produced by the maternal plants exposed to drought [133]. To further utilize maternal effects into breeding programs epigenetic mechanisms of observed maternal effects should be carefully examined.

The main research work has been focused on identification of genetic material that can be utilized in breeding programs. In Italy 21 landraces, one commercial variety and two breeding lines were screened under severe drought stress and favourable conditions through a phenotyping platform that revealed substantial variation in white lupin available genetic material [134]. Two yellow lupin (*L. luteus*) cultivars were tested under drought conditions in order to examine how manipulation of polyamine biosynthesis by using inhibitors of their biosynthesis can influence lupin drought tolerance but the results are not straightforward [135]. Recently several accessions of the Andean lupin (*L. mutabilis*) have been evaluated for their drought tolerance [136] and revealed several favorable traits.

#### 5.3.1. Salt Stress

Few studies have examined salt tolerance in lupin species. A study took place in order to investigate how growth and nitrogen fixation activity was influenced by different NaCl concentrations in a salt-tolerant variety of white lupin *(L. albus*) [137]. It was speculated that salt-tolerance in nodules correlates with a change from osmotic responses to glycoprotein production. Another study was conducted in order to investigate the possibility of utilizing nonpotable water for irrigating horticultural crops. Towards this end, plants from *L. havardii* Wats. (used as a cut flower) and from *L. texensis* Hook. (used as a bedding plant) were drip-irrigated with salt solutions. Results showed that *L. texensis* was more salt-tolerant than *L. havardii*; however, both species were moderately salt tolerant [138]. This genetic material could be utilized in breeding programs with the aim of incorporating salt tolerance to *Lupinus* species used for animal nutrition.

#### 5.3.2. Alkaline Soils

Improvement of white lupin genotypes that can be adapted to high pH values is one of the top priorities for lupin breeding. To this end, lupin landraces from Egypt or Italy that display tolerance to calcareous soils have been identified, and this tolerance can be further enhanced by inoculation with *Bradyrhizobium* strains [30]. Furthermore, sensitivity of white lupin to alkaline soils is a big constraint for its wide distribution in Eastern Europe countries, as well as places where lupin cultivation has recently been introduced. Plant genetic material from the Vavilov collection was tested in Serbia, and the results were quite promising for introgression of these landraces into breeding programs [139]. Another study conducted in Greece evaluated white lupin landraces and advanced populations for yield and tolerance to alkaline soils in order to develop an efficient breeding scheme. Preliminary results by [140] identified valuable genetic diversity among white lupine landraces and advanced populations when selected for both yield and tolerance to alkaline soils.

#### 5.3.3. Breeding Perspectives

It should be noted that the well-organized breeding programs on *L. angustifolius* in Australia [141] highlight the significance of lupin as a high-protein feeding crop with extraordinary adaptation properties explored by breeding for the climate harsh farming systems. Breeding efforts took place in Europe and are currently focused on evaluation and development of lupin cultivars for both human diet and animal feed, to address food safety and the impact of climate change in agriculture.

## 6. Marker-Trait Associations

Detailed genetic studies for the genus Lupinus are not abundant; however, research has ben conducted to identify genes controlling important quality characters. The most critical is low alkaloid content, which enables lupin seed to be used as food or feed. Other ‘domestication syndrome’ characteristics that have been modified include soft, water permeable seeds, which enable easy germination, reduced pod shattering, early flowering, and resistance to anthracnose and phomopsis stem blight (PSB), the most devastating diseases of lupins.

Breeding efforts that begun in 1928/29 in Germany led to the selection of the first low-alkaloid cultivars of *L. angustifolius* [81], and to identification of the genes that cause high alkaloid content in seeds. Quinolizidine alkaloids (QA) in lupins are toxic, bitter secondary metabolites, synthesized in aerial tissues and transported via the phloem to seeds. These compounds provide insect resistance, and total alkaloid content varies from 0.02 to 12.73% of the seed dry weight [142]. QA biosynthesis begins by the decarboxylation of lysine to produce cadaverine, which is further modified by various reactions (Figure 2).

In this process only three genes/enzymes, the *Lupinus angustifolius* lysine/ornithine decarboxylase (LaL/ODC), *Lupinus angustifolius* copper amine oxidase (LaCAO), and *Lupinus albus* tigloyl-CoA:(−)-13α-hydroxymultiflorine/(+)-13α-hydroxylupanine O-tigloyltransferase (LaHMT/HLT) have been characterized. The action of other gene/enzymes such as p-coumaroyl-CoA/feruloyl-CoA:(+)-epilupinine/(−)-lupinine O-coumaroyl/feruloyltransferase (ECT/EFT-LCT/LFT) and other acetyl transferases (AT), berberine bridge enzyme (BBE) and major latex-like proteins (MLP-like), which were more highly expressed in leaf tissue of bitter versus sweet NLL, have been postulated [143,144].

Several mutations that cause lower alkaloid levels in sweet lupin varieties for the different cultivated lupin species have been identified in various breeding efforts [124]. None of them involves the three characterized genes, and for two of them (*iucundus* and *pauper*), markers have been developed to enable MAS. Significant effort was devoted to developing a marker for the recessive mutant at the *Iucundus* locus of NLL, which reduces alkaloids from ‘bitter’ wild lupin to ‘sweet’ cultivars. All lupin cultivars in Australia are homozygous for the recessive allele. *Iucundus* mutation is difficult to track in breeding programs. The gene locus has been maped at the linkage group 7 of NLL in a gene-rich region where 26 genes have been identified [145,146]. This region harbors regulatory genes and those possibly involved in alkaloid biosynthesis [144]. Li et al. [147] developed the marker IucLi, located 0.9 cM away from *iucundus*. The accuracy between marker genotype and phenotype was 100% in the 25 common cultivars and 86.4% among the 125 accessions of NLL lupin core collection examined. Sweet NLL that has the iuc mutation is characterized by reduced levels of the first enzymes of the QA pathway (LaL/ODC and LaCAO) compared with bitter NLL, suggesting that this gene may act in the regulation of the QA biosynthetic pathway (Figure 2, [124,144]).

In *L. albus* (white lupin, WL), all old traditional varieties are bitter. Breeding efforts in the early 20th century resulted in the identification of sweet WL mutants harboring mutations at the *pauper*, *exiguus* and *nutricius* genes, all of which were incorporated in commercial varieties. Genetic studies indicated that these three low-alkaloid genes are not linked [148]. The *pauper* locus is the most effective mutation in reducing QA levels, and is now used almost exclusively in breeding programs [124]. The low-alkaloid loci *dulcis* of *L. luteus* and presumably *pauper* in *L. albus* do not affect lysine and cadaverine levels, as these do not differ between sweet and bitter cultivars, and nor do they affect activities of QA acyltransferases, suggesting that mutations may act immediately downstream of cadaverin in the QA biosynthetic pathway (Figure 2, [149]). Lin et al. [150] developed a PCR-based marker PauperM1, closely linked to the gene *pauper* at 1.4 cM, which gives high accuracy in selection of the pauper genotype when it is used for MAS in lupin breeding.

Lupins are known as being ‘hard-seeded’ because they produce seeds that resist water imbibition, preventing germination of every wetted seed. This adaptation is beneficial in the wild, because it allows for a seed reservoir in the soil to be used to maintain the species after a disastrous year that produced low seed, but is unwanted by farmers, who desire high germination rates and uniform plant densities. A single recessive gene *mollis* in NLL with unknown action results in seeds with water-permeable testa, and markers for the mutant genotype have been developed [151,152]. A co-dominant, sequence-specific PCR marker, designated MoLi, showed a consistent match with phenotypes of seed coat permeability in 25 cultivars, and 119 of the 125 mollis genotypes of the accessions in the Australian *L. angustifolius* core collection [152]. Thus, marker MoLi provides the means to select the *mollis* gene in a wide range of NLL genotypes in lupin breeding.

Lupin is a long-day plant, and its flowering time can be affected by prolonged exposure to cold, ambient temperatures, and photoperiod [153]. Wild genotypes are generally late-flowering and require vernalization; however, domesticated NLL cultivars carry the early flowering dominant gene *Ku*, which removes this requirement. Early flowering is also an important characteristic for adaptation to short growing seasons. A co-dominant sequence-specific marker KuHM1 was developed, which showed perfect correlation between plant phenotype and the marker score on a RIL population, giving a distance estimate of <0.5 cM from the gene *Ku* [151]. Yet, to be useful for MAS, this marker should be confirmed in a broad range of NLL genotypes, and the precise genetic distance from *Ku* would require further testing on a larger population. In searching for the identity of the gene underlying the *Ku* locus, Nelson et al. [73] presented evidence that a large deletion in the 5′ regulatory region of an *Flowering Locus T* (*FT*) homologue is associated with the loss of vernalization responsiveness at the *Ku* locus. Kroc et al. [154] developed three *FT* homologue-based markers, one of which (‘dFTc’) mapped to the same genetic position as the *Ku* locus and warranted further investigation as a potential candidate gene for *Ku*. It should be noted that FT is the major signal for transition from vegetative to inflorescence development. Genotyping using a PCR marker for the 1.4-kb indel in the 5′ flanking sequence of FTc1 confirmed its expected co-location with the *Ku* locus, and phenotype–genotype correlation between indel variation and vernalization responsiveness was 100% in a panel of 50 domesticated cultivars and 166 wild accessions [73]. This suggests that a functional marker system to access early flowering genotypes at the *Ku* locus is available for MAS.

Pod shattering is another “domestication syndrome” characteristic that is present in wild lupins and should be removed for development of modern cultivars. The characteristic is controlled by the genes *tardus* and *lentus*, each of which partially confer reduced pod shattering, and only plants harboring both genes are non-shattering Glastones [155]. The gene *tardus* limits shattering for the account of accretion of beans valves by the formation of solid beam of sclerenchymal cell on all perimeters of a bean, while the gene *lentus* reduces shattering at the expense of structural changes in the valves [156]. Boersma et al. [157] developed three locus-specific markers flanking the *tardus* gene that showed moderate phenotype association with marker genotypes (24–39%) on a set of 33 wild accessions tested. Later, Li et al. [158] developed TaLi, a PCR-based co-dominant marker mapped 1.4 cM from the *tardus* gene, which displayed a 94% match of phenotype and genotype marker for 25 domesticated commercial cultivars and 125 accessions of the lupin core collection. For molecular identification of *lentus* genotypes, Boersma et al. [151] developed the markers LeM1 and LeM2, flanking the gene on the same side at 2.6 cM and 1.3 cM, respectively, which showed a correlation of 95% with the plant phenotype for the Australian cultivars, and approximately 36% on wild types tested. Another co-dominant DNA marker linked to the gene *lentus* was developed by Li et al. [159]. The marker, designated LeLi, was applied to 25 cultivars released in Australia and 125 wild core accessions of the Australian Lupin Collection, scoring an overall matching rate of 60.67%, assuming correct determination of the phenotypes [159].

Anthracnose caused by *Colletotrichum gloeosporioides* (Penz.) is the most devastating disease of lupins worldwide. A marker AntjM1, tagging the anthracnose disease resistance gene *Lanr1*, was developed [103]. Unfortunately, cultivars highly susceptible to the disease were found to have the resistance marker allele. Because of the importance of resistance to anthracnose disease for release of new lupin cultivars, continuing research efforts led to the development of the diagnostic marker AntjM2 for the *Lanr1* gene [95]. As reported by Yang et al. [160], application of molecular markers to diagnose anthracnose resistance/susceptibility enabled selection of 98% of breeding lines resistant to the disease, as compared to 50–60 % of advanced breeding lines being thrown away due to the disease before application of MAS. An overview of the molecular markers developed to incorporate specific genetic loci and traits in lupin breeding programs is given in Table 2.

Another devastating disease for lupins is phomopsis stem blight disease caused by the fungal pathogen *Diaporthe toxica* (formerly called *Phomopsis leptostromiformis*). A *PSB resistance* locus was identified in a recombinant inbred line (RIL) population derived from the cross of Tanjil (resistant to PSB) and Unicrop (susceptible) using next-generation sequencing (NGS)-based restriction site-associated DNA sequencing [161]. Thirty-three SNP markers showed correlation between the marker genotypes and the PSB disease phenotype and seven of them were converted into sequence-specific PCR markers, all located at 2.1 cM from the resistance locus. Two of them, PhtjM4 and PhtjM7 are recommended in MAS for PSB resistance in the Australian national lupin breeding program due to its wide applicability on breeding germplasm and close linkage to the putative resistance gene [109].

With recent advances in NGS technologies, the development of sequence-based markers for application in MAS has become easier and more cost effective. New variants can be identified by sequencing cultivars and wild accessions, and sequence-specific primers can be designed to screen the variants using cost-effective techniques like high-resolution melting (HRM) or developing primers for specific sequencing applications such as the Ampliseq^TM^ technology (ThermoFisher Scientific). In such an effort, the draft genome sequence of NLL was obtained, and each of the five ‘domestication syndrome’ genes (*Iucundus*, *Mollis*, *Tardus*, *Lentus*, *Ku*), as well as other previously established markers, were converted into SNP markers which are now applied for MAS in the F2 populations [121]. Lupin is a species with a relatively small genome, and reference genome assemblies have been obtained for NLL and WL (see below). The available sequence information enables further screening for identification of new Quantitative Trait Locus (QTLs) and development of diagnostic functional markers that will accelerate breeding efforts for new cultivars with improved agricultural traits.

## 7. Genome Sequence Efforts in Lupins

The recent sequencing of many complete Fabaceae genomes, including those of white and narrow-leaved lupins, along with the availability of high-throughput resources, offers new challenges and opportunities for lupin breeders in the genomic era towards the improvement of climate-smart traits and disease resistance against the main phytopathogenic organisms that plague crops. Genomics technologies will certainly generate a large amount of data and lead to modern breeding objectives for the production of lupins that are nutritious, meet the dietary preferences in animal feeding.

*L. angustifolius* L., (NLL), is an important grain legume crop that is valuable for sustainable farming and is recognized as a potential human health food. Recent interest is being directed at NLL to improve grain production, disease and pest management and health benefits of the grain. However, studies have been hindered by a lack of extensive genomic resources for the species. Gao et al. [162] constructed a NLL Bacterial Artificial Chromosome (BAC) library consisting of 111,360 clones with an average insert size of 99.7 Kbp from cv. Tanjil. The library has approximately 12× genome coverage. Both ends of 9600 randomly selected BAC clones were sequenced to generate 13985 BAC end-sequences (BESs), covering approximately 1% of the NLL genome. These BESs permitted a preliminary characterization of the NLL genome, including aspects such as organization and composition, with the BESs having approximately 39% G:C content, 16.6% repetitive DNA, and 5.4% putative gene-encoding regions. They identified 9966 simple sequence repeat (SSR) motifs from the BESs and found some of these to be potential markers. They suggested that the NLL BAC library and BAC-end sequences are powerful resources for genetic and genomic research on lupin. According to Hong et al. [163], each BAC is expected to overlap with enough other BACs to provide a solid framework for assembling long-range scaffolds of genomic sequences. Thus, in this way we could have the complete sequence map of the lupin genome. Another study, by Yang et al. [161], reported the draft assembly from a whole genome shotgun sequencing dataset for *L. angustifolius* with 26.9× coverage of the genome, which is predicted to contain 57,807 genes. They suggested that the number of genes identified was greater than that found in other legume species. Recently, Hane et al. [145] generated a high-quality reference genome assembly (609 Mb), which captured >98% of the gene content. Additionally, they generated in-depth RNAseq datasets from five different tissue types, being roots, stems, leaves, flowers and seeds [164]. They used these datasets to develop gene-based molecular insertion/deletion (indel) and SNP markers. Current research focuses on the generation of a pan-genome for the species using forty genetically different NLL accessions [165]. These resources have led to the identification of candidate genes for several crucial characteristics. Hence, the developed resources will significantly improve and accelerate NLL breeding programs, especially since NLL has only been ‘domesticated’ over the last 50 years.

Another lupin species, white lupin (*L. albus*; 2*n* = 50), stands out as a model legume species, since it is the only crop producing cluster roots, one of the most outstanding developmental adaptations to nutrient-scarce environments. Hufnagel et al. [166] reported a high-quality chromosome-scale assembly of the white lupin genome, together with extensive transcriptome data from ten different tissues. They used single-molecule real-time technology, in combination with short-read sequencing and optical and genetic maps, for assembly. They found that the final assembly size was 451 Mb, with a N50 of 17 Mb. About 96% (434 Mb) of the assembled genome is included on the 25 pseudo-chromosomes. The structural annotation identified 38,258 coding genes and 3,129 ncRNA, with 97.3% genes being anchored on the pseudo-chromosomes. Several transcriptomic and proteomic studies have been accomplished on roots of white lupin in an attempt to unravel its acclimation strategy to soils with high concentrations of heavy metals [167] or with low phosphorous concentration [168]. In particular, O’Rourke et al. [168] carried out an RNA-Seq whole-transcriptome analysis in order to unravel unique transcript sequences (more than 2000), highly responsive to Pi deficiency in white lupin, and mapped them to BAC end sequences of narrow-leaved lupin. Furthermore, the white lupin genome consists of gene duplications and repetitive elements and presents extensive duplication blocks inside its own genome; thus, it is a valuable resource and represents a keystone for legume genomics research. These two draft genomes will support further whole-genome analysis of other species of Papilionoideae legumes.

## 8. The Use of Genomic Tools in Breeding Programs

The narrow-leaved lupin genome and the white lupin genome are the most studied genomes for the *Lupinus* genus; however, the first one is considered to be the reference genome as several genome maps have been integrated and utilized for comparative mapping within model legume plants [169]. The available genomic tools are listed in Table 3. The first linkage map for white lupin was constructed using microsatellite-anchored fragment length polymorphism (MFLP), and DNA markers were generated that were tightly linked to domestication genes such as low alkaloid content, reduced pod shattering, etc. [170]. Later on, based on HRM-STS (sequence target sites) markers, comparisons between the genetic maps of white lupin and narrow-leaved lupin were made possible [171]. Nevertheless, the linkage map of narrow-leaved lupin that was generated from the segregation of 135 gene-based PCR markers [172] enabled cross-species analysis, resulting in the detection of conserved synteny between *L. angustifolius* and *Medicago truncatula*, and later to the unification of the genetic map of these species and its alignment to the genome sequence of *Lotus japonicus* [146]. Moreover, the genome of narrow-leaved lupin contains gene-rich regions that obtain a high degree of synteny with another model legume, *Glycine max* [173].

A high-density linkage map for white lupin was created recently in order to identify QTLs carrying genes of agronomic importance such as seed alkaloid content, vernalization requirements, and resistance to Phomopsis stem blight and anthracnose [174]. Furthermore, they validated PCR-markers in order to map these QTLs and utilize them in marker-assisted selection for lupin breeding. Previous work on identification of QTLs for anthracnose resistance and flowering time was accomplished by [93]. However, due to several limitations that QTL approaches for lupin breeding have encountered, association mapping approaches that take into account the population structure of the mapping species are becoming very popular [24]. To this end, Iqbal et al. [175] tested a total of 122 accessions of white lupin by using association mapping with the Amplified fragment length polymorphism (AFLP) markers for seed weight, a highly significant agronomic trait. Two markers were indentified that explained 22.69% and 20.5% of seed weight variation, respectively, suggesting the suitability of the aforementioned germplasm for association mapping studies.

Interestingly, Książkiewicz et al. [174] came up with a very important finding. Although they discovered high collinearity between white lupin and narrow-leaved lupin genomes, as indicated by other research groups as well [166], they showed that some traits with great agronomic importance (i.e., early flowering and anthracnose resistance) are controlled by different regions in the aforementioned genomes and by different number of genes that are inherited in a different manner. Thus, breeding of white lupin should rely mostly on information provided by its own genome sequencing [174].

## 9. Conclusions

This study provides information about lupin species that can be utilized for animal feeding. Creating a suitable biological material (genotypes/cultivars) and maximizing its yield and productivity would represent an increase in the economic value of lupin cultivation and raw material processing. Modern targets in lupin breeding include yield stabilization, resistance to biotic and abiotic stresses, biochemical structure associated with seed quality, and late maturing. In particular, seed quality features are of great importance for animal feeding. Up to now, the breeding efforts have substantially contributed to reducing the alkaloid content, and modern varieties contain low levels of alkaloids. In the future, breeding has to be focused on the improvement of other qualitative characteristics of the seed, such as the content of non-starch polysaccharides and oligosaccharides. The sources of genetic variation used to achieve the aforementioned breeding targets include landraces, wild species, gene bank accessions and new variants derived after technical and in vitro mutagenesis.

Evolutionary studies combining genetic and spatial eco-climatic datasets are of immense importance to unraveling genes and/or genetic loci that natural selection has settled. This information can be used in breeding programs to identify target groups to harness the undisclosed genetic diversity. Implementation of marker-assisted selection or genomic selection, for improving quantitative traits using a combination of phenotyping and genotypic data, is of great significance for future breeding efforts. Genomic technologies along with phenotyping approaches would revolutionize our understanding and assist in developing strategies for both conservation and utilization. This review conclusively highlights breeding targets of lupin as a legume crop to address the adaptability, resilience and sustainability of agro-ecosystems in a changing environment. Thus, it can be said that lupine is a full-featured alternative to soybean with the advantage of a relatively strong prebiotics effect.

## Figures and Tables

**Figure 1 ijms-20-00851-f001:**
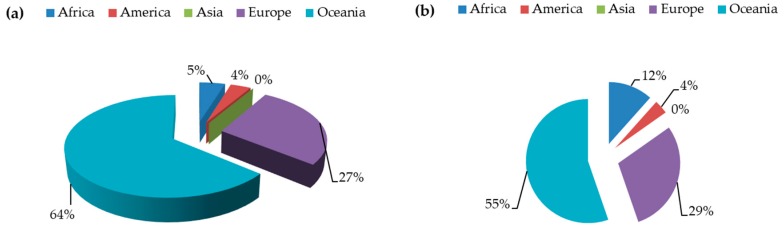
Worldwide distribution of (**a**) lupin production and (**b**) cultivated area. Source: FAOSTAT2018.

**Figure 2 ijms-20-00851-f002:**
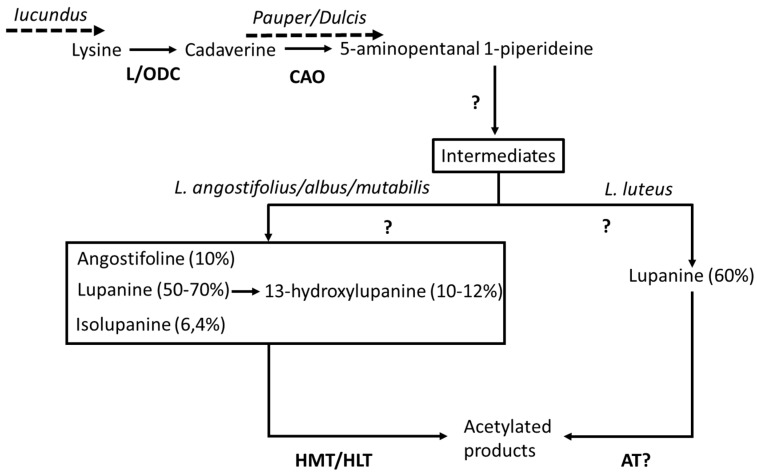
A simplified scheme of the QA biosynthetic pathway in lupin species. Known gene/enzymes and the major alkaloids with concentrations in parentheses are indicated. Genetic loci with characterized mutants resulting in lower alkaloid concentrations are shown above dashed arrows indicating the step of the pathway that is affected by the mutation. Question marks indicate unknown steps in the pathway. LDC, Lysine/ornithine decarboxylase; CAO, Copper amine oxidase; HMT/HLT, tigloyl-CoA: (−)-13α-hydroxymultiflorine/(+)-13α-hydroxylupanine O-tigloyltransferase; AT, Acetyltransferase.

**Table 1 ijms-20-00851-t001:** Chemical composition of the cultivated lupin species.

	*L. albus ^a^*	*L. angustifolius ^b^*	*L. luteus ^c^*	*L. mutabilis ^d^*
Crude Protein (% of DM)	33–47	31–37	37–38	32–52
Crude fibre (% of DM)	13–16	15–17	12–15	10
Metabolized Energy (MJ/kg DM)	13–16	12–13	10	na
Oil (%)	6–13	6–7	5–9	13–24
Total oligosaccharides (% of DM)	7–8	8–9	na	na
Non-starch polysaccharides (%)	18	47–51	na	na
**Essential amino acids (g/16 gN)**				
Lys	4.9–5.1	4.5–5.0	4.2–4.6	5.0–7.3
Met	0.6–0.7	0.6–0.7	0.6–0.7	0.4–1.4
Cys	1.8–2.1	1.3–1.6	1.8–2.5	1.4–1.7
Leu	7.5–8.0	6.0–7.6	6.1–7.3	5.7–7.8
Thr	3.1–4.0	3.0–3.3	2.6–3.2	3.0–4.0

^a^ [13,42,45,46]; ^b^ [36,47]; ^c^ [44,46,48]; ^d^ [12,49,50]; na: not available.

**Table 2 ijms-20-00851-t002:** Molecular markers developed and used in lupin breeding programs.

LOCUS	Trait	Species ^a^	Marker	Ref.
*Iucundus*	low alkaloids	NLL	IucLi	[147]
*pauper*	low alkaloids	WL	PauperM1	[150]
*mollis*	water-permeability of testa	NLL	MoLi	[152]
*Ku*	Early flowering—vernalization	NLL	KuHM1	[151]
*Ku*	Early flowering—vernalization	NLL	dFTc	[154]
*tardus*	reduced shattering	NLL	TaLi	[158]
*lentus*	reduced shattering	NLL	LeM1, LeM2	[151]
*lentus*	reduced shattering	NLL	LeLi	[159]
*Lanr1*	anthracnose disease resistance	NLL	AntjM2	[95]
*PSB-res*	phomopsis stem blight resistance	NLL	PhtjM4, PhtjM7	[109]

^a^ NLL: *L. angustifolius*, WL: *L. albus*.

**Table 3 ijms-20-00851-t003:** List of available genomic tools in *Lupinus albus* and *Lupinus angustifolius*.

*Lupinus albus* L.			*Lupinus angustifolius* L.		
BAC libraries		References	BAC libraries	Two centromeric BAC clones; 111, 360 clones (12× coverage); (26.9× coverage containing 57,807 genes)	[161,162,169]
Genome assemblies			Genome assemblies	Draft genome sequence (609 Mb)	[145]
Genetic linkage maps	MFLP loci; HRM-STS markers	[170,171]	Genetic linkage maps	STS markers	[146]
Transcriptome and proteome assemblies	De Novo Transcriptome assembly, gene annotation and functional classification; EST and protein datasets	[167,168]	Transcriptome and proteome assemblies	335 transcriptome-derived markers	[164]
QTL mapping	seed alkaloid content, flowering time, resistance to Phomopsis stem blight, anthracnose resistance	[93,174]	QTL mapping		
Association mapping	AFLP markers for seed weight	[175]	Association mapping		
Comparative mapping	*Lupinus angustifolius*; *Medicago truncatula*	[166,174]	Comparative mapping	*Lotus japonicus*; synteny analysis among legume species; *Glycine max*	[146,169,173]

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
