# Peer review of "The Use of Lupin as a Source of Protein in Animal Feeding: Genomic Tools and Breeding Approaches"

_ijms, 2019, doi:10.3390/ijms20040851_

Round 1
Reviewer 1 Report
This manuscript is a very detailed and up-to-date description of Lupin use in agriculture as source of protein in animal feeding. The manuscript is very well written and well-organized: starting with a geo-economical overview of lupins and their role in the farming systems. The available genomic resources and the current breeding methods are extensively described. Finally, the new opportunities offered by the genome of Lupins and the development of genomic tools in breeding programs are presented and discussed.
I think this manuscript offers a comprehensive and objective vision of Lupins research, focusing on the breeding programs.
The authors did an impressive job collecting and describing such a large amount of information, which will surely be of great value for all those working in Lupins research and breeding programs. I understand, and appreciate, that authors gave a very neutral and unbiased vision of the advances in Lupins research, sometimes proposing what they think it’ll be important in the near future. I would have enjoyed getting their opinion/feeling on what they think will be the solutions to address the critical issues they clearly identified and described.
I think the manuscript reads well and I have only few suggestions and doubts to improve it.
My main doubt is that, on one hand, lupin alkaloid levels are a primary antinutritional factors limiting lupin use in animal feeding. On the other hand, alkaloids are essential to trigger proper defenses in lupin. Do you think it would be possible to uncouple alkaloid antinutritional and defense properties? Which strategies are being employed to reduce alkaloid antinutritional without affecting alkaloid-promoted defenses? According to what I understand, conventional breeding and germoplasm diversity are not the right strategies, because they’ll identify traits that regulate alkaloid levels without uncoupling antinutritional and defense properties. Any suggestions on how this issue is being (or should be) addressed? For example, is the generation/selection of high alkaloid in leaves (to promote plant defence) and alkaloid-free seeds (to reduce antinutrition) varieties a current strategy? Is the leave-to-seed transport of alkaloids a currently studied trait?
In addition, the abstract is clear and easy to read. The problem to address and main objectives of this review are well-defined.
However, I think there is a contradictory sentence in the Abstract “the presence of quinolizidine alkaloids and their specific carbohydrate composition are the main antinutritional factors that prevent their use in animal feeding” compare to the text (line 152) “the alkaloids’ level of current cultivars are very low [44] and they do not affect feed intake”. It is not clear if alkaloids in lupins are a still a problem or not. Please, make it clear and consistent.
Small points to be addressed:
1) what CAP (line 54) stand for ?
2) could you add a sentence or two describing the grain legumes, other than lupin, that are currently used as alternative source of protein in animal feed?
3) line 140, why is the low content of methionine the main problem of their use as animal feed?
4) line 141 “oil content of lupins is generally quite low” compared to what? other grain legumes?
5) line 145 “high-quality oil profile for the grains of both L. albus and L. luteus” compared to what? other grain legumes?
6) line 336, the parenthesis is open (, but not closed.
7) Authors described different strategies to get lupin resistance or tolerant to different diseases. However, in the case of Fusarium oxysporum, that is a very important fungal pathogen, no strategies to get tolerant lupin are described. It’ll be nice to review the current strategies to improve lupin tolerance to Fusarium, even if there are not based on breeding r resistance genes.
8) paragraph in line 442-447 is a general conclusion on the breeding programs and it does not fit well within the “5.3.2. Alkaline soils” section. Shall it be moved somewhere else or given an individual title?
9) what about including a Table with all the genes/loci affected in described in the “Marker-Trait Associations” text? It can be a very useful and visual summary of the text.
10) line 599 “They suggested that the NLL BAC library and BAC-end sequences are powerful resources for genetic and genomic research on lupin.” Why are they powerful tools? Did they indentify markers linked to disease resistance or stress tolerant traits? Please describe a bit more in details how this genomic information can be used in improving the use of lupin.
11) can you finish the “7. Genome Sequence Efforts in Lupins” section with a general take home message about lupins’ genome? Maybe a comparison with the available high-resolution legumes/fabaceae genomes?
12) line 651 “Towards this end, Iqbal et al. [164] tested a total of 122 accessions of white lupin by using association mapping with the AFLP markers for seed weight, a highly significant agronomic trait.” What was the outcome of this research? Did they identify any markers linked to this agronomic trait? Please summarize the main result of this analysis.
13) there are a few sentences in the “Conclusions” that are difficult to understand. Can you revise and re-phrase the first two sentences (line 663-665) and that in line 674-5, avoiding colloquial words such as “sorted out”?
14) please add the full online link of the source, including the doi in case it is available for references 3, 4, 7 and 25.
15) Ref 11 is from 1992, are there any more recent research on higher monogastric susceptibility to anti-nutritional factors compared to ruminants?
Author Response
Response to Reviewer 1 Comments
Thank you for your comments and suggestions on our manuscript. We are pleased that the reviewers have positive feedback on manuscript. Please see below for our responses to
your comments and suggestions. In the revised manuscript, all changes are distinguished by track changes.
Point 1: My main doubt is that, on one hand, lupin alkaloid levels are a primary antinutritional factors limiting lupin use in animal feeding. On the other hand, alkaloids are essential to trigger proper defenses in lupin. Do you think it would be possible to uncouple alkaloid antinutritional and defense properties? Which strategies are being employed to reduce alkaloid antinutritional without affecting alkaloid-promoted defenses? According to what I understand, conventional breeding and germoplasm diversity are not the right strategies, because they’ll identify traits that regulate alkaloid levels without uncoupling antinutritional and defense properties. Any suggestions on how this issue is being (or should be) addressed? For example, is the generation/selection of high alkaloid in leaves (to promote plant defence) and alkaloid-free seeds (to reduce antinutrition) varieties a current strategy? Is the leave-to-seed transport of alkaloids a currently studied trait?
Response 1: Thank you for the comment. We were missing this matter. Yes, it is true the cultivars with low alkaloid level are more susceptible especially to the aphids. However, not only the alkaloid level but also the alkaloid profile is of great importance for tolerance to aphids. Thus, the breeding efforts have to be focused on a specific alkaloid profile that contribute to the resistance and in addition to retain the level of alkaloids to leaves and to reduce it in seeds. A paragraph about this has been added to text.
Point 2: However, I think there is a contradictory sentence in the Abstract “the presence of quinolizidine alkaloids and their specific carbohydrate composition are the main antinutritional factors that prevent their use in animal feeding” compare to the text (line 152) “the alkaloids’ level of current cultivars are very low [44] and they do not affect feed intake”. It is not clear if alkaloids in lupins are a still a problem or not. Please, make it clear and consistent
Response 2: The modern cultivars of L. albus and L. angustifolius with commercial use are almost free of alkaloids. A sentence has been added to the abstract to clarify.
Point 3: what CAP (line 54) stand for ?.
Response 3: Done as requested.
Point 4: could you add a sentence or two describing the grain legumes, other than lupin, that are currently used as alternative source of protein in animal feed?
Response 4: Done as requested.
Point 5: line 140, why is the low content of methionine the main problem of their use as animal feed?
Response 5: Because Methionine is a nutritionally essential, sulfur-containing amino acid. The other sulfur-containing amino acid that limits the nutritional quality of plants is cysteine. However, methionine can be converted into cysteine in human and animal cells. Thus it is important element of the diet of monogastric. We rewrote the sentence.
Point 6: line 141 “oil content of lupins is generally quite low” compared to what? other grain legumes?
Response 6: Compare to oilseeds. It was added to the text.
Point 7: line 145 “high-quality oil profile for the grains of both L. albus and L. luteus” compared to what? other grain legumes?
Response 7: Compared to other vegetable oils. It was added to the text.
Point 8: line 336, the parenthesis is open (, but not closed.
Response 8: Done as requested.
Point 9: Authors described different strategies to get lupin resistance or tolerant to different diseases. However, in the case of Fusarium oxysporum, that is a very important fungal pathogen, no strategies to get tolerant lupin are described. It’ll be nice to review the current strategies to improve lupin tolerance to Fusarium, even if there are not based on breeding r resistance genes
Response 9: The current strategies to improve lupin tolerance to Fusarium have been added
Point 10: paragraph in line 442-447 is a general conclusion on the breeding programs and it does not fit well within the “5.3.2. Alkaline soils” section. Shall it be moved somewhere else or given an individual title?
Response 10: Done as requested.
Point 11: what about including a Table with all the genes/loci affected in described in the “Marker-Trait Associations” text? It can be a very useful and visual summary of the text.
Response 11: Done as requested.
Point 12: line 599 “They suggested that the NLL BAC library and BAC-end sequences are powerful resources for genetic and genomic research on lupin.” Why are they powerful tools? Did they indentify markers linked to disease resistance or stress tolerant traits? Please describe a bit more in details how this genomic information can be used in improving the use of lupin.
Response 12: Large-insert genomic DNA libraries in bacteria, such as bacterial artificial chromosome provides a way to divide the complexity of the lupin genome into a composite of large DNA segments of reduced complexity. BAC end sequences can serve as sequence-tagged sites (STSs) for a whole-genome shotgun sequencing project. Especially, BAC-end pair mates obtained from a BAC library containing 10–15 genome equivalents are useful because they can provide long-range (100 kb) sequence information for the assembly of shotgun sequenced DNA fragments (i.e., 2-kb and 10-kb plasmid inserts) into long-range scaffolds.
We added these sentence ‘According to Hong et al. (2003), each BAC is expected to overlap with enough other BACs to provide a solid framework for assembling long-range scaffolds of genomic sequence. Thus, with this way we could have the complete sequence map of the lupin genome’
Point 13: can you finish the “7. Genome Sequence Efforts in Lupins” section with a general take home message about lupins’ genome? Maybe a comparison with the available high-resolution legumes/fabaceae genomes
Response 13: We added this sentence ‘These two draft genomes will support further whole‐genome analysis of other species of Papilionoideae legumes’ in the revised manuscript.
Point 14: line 651 “Towards this end, Iqbal et al. [164] tested a total of 122 accessions of white lupin by using association mapping with the AFLP markers for seed weight, a highly significant agronomic trait.” What was the outcome of this research? Did they identify any markers linked to this agronomic trait? Please summarize the main result of this analysis.
Response 14: Done as requested.
Point 15: there are a few sentences in the “Conclusions” that are difficult to understand. Can you revise and re-phrase the first two sentences (line 663-665) and that in line 674-5, avoiding colloquial words such as “sorted out”?
Response 15: Done as requested.
Point 16: please add the full online link of the source, including the doi in case it is available for references 3, 4, 7 and 25.
Response 16: Done as requested.
Point 17: Ref 11 is from 1992, are there any more recent research on higher monogastric susceptibility to anti-nutritional factors compared to ruminants?
Response 17: Done as requested.
Reviewer 2 Report
I reviewed the manuscript, the authors provided an alternative of soybean using Lupin as the protein source in animal feeding. This manuscript was well organized. However, before publishing, a few questions need to be figured out.
In page 2 line 54, please clarify what is the CAP indicated?
According to the description on page 3 and line 120-128, can authors summary what factors limited the production of Lupin in Europe?
As an alternative crop for aminal feed, the yield should be very important, can authors include more yield information? Such as research on yield-related traits, the comparison between soybean and Lupin.
Author Response
Response to Reviewer 2 Comments
Thank you for your comments and suggestions on our manuscript. We are pleased that the reviewers have positive feedback on manuscript. Please see below for our responses to your comments and suggestions. In the revised manuscript, all changes are distinguished by track changes.
Point 1: In page 2 line 54, please clarify what is the CAP indicated?.
Response 1: Done as requested.
Point 2: According to the description on page 3 and line 120-128, can authors summary what factors limited the production of Lupin in Europe?
Response 2: Done as requested.
Point 3: As an alternative crop for aminal feed, the yield should be very important, can authors include more yield information? Such as research on yield-related traits, the comparison between soybean and Lupin
Response 3: Done as requested.